# In Search of Covariates of HIV-1 Subtype B Spread in the United States—A Cautionary Tale of Large-Scale Bayesian Phylogeography

**DOI:** 10.3390/v12020182

**Published:** 2020-02-05

**Authors:** Samuel L. Hong, Simon Dellicour, Bram Vrancken, Marc A. Suchard, Michael T. Pyne, David R. Hillyard, Philippe Lemey, Guy Baele

**Affiliations:** 1Department of Microbiology, Immunology and Transplantation, Rega Institute for Medical Research, KU Leuven, 3000 Leuven, Belgium; simon.dellicour@kuleuven.be (S.D.); bram.vrancken@kuleuven.be (B.V.); philippe.lemey@kuleuven.be (P.L.); guy.baele@kuleuven.be (G.B.); 2Spatial Epidemiology Lab (SpELL), Université Libre de Bruxelles, 1050 Brussels, Belgium; 3Department of Biomathematics, David Geffen School of Medicine at UCLA, University of California, Los Angeles, CA 90095, USA; msuchard@ucla.edu; 4Department of Human Genetics, David Geffen School of Medicine at UCLA, University of California, Los Angeles, CA 90095, USA; 5Department of Biostatistics, Fielding School of Public Health, University of California, Los Angeles, CA 90095, USA; 6ARUP Institute for Clinical and Experimental Pathology, Salt Lake City, UT 84108, USA; michael.pyne@aruplab.com; 7Department of Pathology, University of Utah, Salt Lake City, UT 84112, USA; hillyadr@aruplab.com

**Keywords:** HIV, BEAST, BEAGLE, phylogeography, phylodynamics, spread, covariates, predictors

## Abstract

Infections with HIV-1 group M subtype B viruses account for the majority of the HIV epidemic in the Western world. Phylogeographic studies have placed the introduction of subtype B in the United States in New York around 1970, where it grew into a major source of spread. Currently, it is estimated that over one million people are living with HIV in the US and that most are infected with subtype B variants. Here, we aim to identify the drivers of HIV-1 subtype B dispersal in the United States by analyzing a collection of 23,588 *p*ol sequences, collected for drug resistance testing from 45 states during 2004–2011. To this end, we introduce a workflow to reduce this large collection of data to more computationally-manageable sample sizes and apply the BEAST framework to test which covariates associate with the spread of HIV-1 across state borders. Our results show that we are able to consistently identify certain predictors of spread under reasonable run times across datasets of up to 10,000 sequences. However, the general lack of phylogenetic structure and the high uncertainty associated with HIV trees make it difficult to interpret the epidemiological relevance of the drivers of spread we are able to identify. While the workflow we present here could be applied to other virus datasets of a similar scale, the characteristic star-like shape of HIV-1 phylogenies poses a serious obstacle to reconstructing a detailed evolutionary and spatial history for HIV-1 subtype B in the US.

## 1. Introduction

Next-generation sequencing technologies have introduced quicker and cheaper sequencing, and have ushered in a new era of research in infectious disease genomics [1,2]. The result has been an explosion in the availability of densely-sampled pathogen populations, which has made it possible for researchers to study disease transmission at an unprecedented level of resolution, from the host-to-host to the within-host scales [3,4]. It is within this context that Bayesian phylodynamics has risen to become a popular framework to infer epidemiological relations between pathogen genomes. Phylodynamics aims to “clarify how pathogen genetic variation, modulated by host immunity, transmission bottlenecks, and epidemic dynamics, determines the wide variety of pathogen phylogenies observed at scales that range from individual host to population” [5]. A specific type of phylodynamic analysis that can be performed is virus phylogeography [6,7]. Virus phylogeography aims to answer the question of “how did an epidemic spread through space and time?" by connecting the evolutionary and geographic dispersal processes. For rapidly evolving viruses and—thanks to the ease by which sequencing technologies nowadays allow the generation of (near) complete genomes—also for slower-evolving pathogens, these processes are measurable within the same time scale [8].

Bayesian phylogeographic inference using trait diffusion models aims to achieve this by jointly reconstructing the evolutionary and geographical history of a virus population and estimating the locations of ancestral viruses represented by internal nodes on a phylogenetic tree conditioned on the observed locations at the tips. Discrete locations can be modeled as discrete traits using continuous-time Markov chain models [9]—analogous to nucleotide substitution modeling—while continuous locations make use of Brownian diffusion models of continuous trait evolution, allowing for branch-specific dispersal rates through the use of random-walk models [10]. Both discrete and continuous phylogeographic models have been extended to incorporate epidemiological and ecological data, enabling the testing of hypotheses concerning their effect on spatial spread [11,12].

The different trait diffusion models come with different assumptions and their application may also depend on the spatial resolution of the location data. The discrete model makes abstractions of precise geographic data—sometimes made necessary by the absence of fine-scale geographic information—but may still provide a powerful approach to infer epidemiological dynamics, as illustrated in applications like investigating the relevance of migration on public health planning [13,14,15,16,17], identifying the determinants of the spread of dog rabies virus in North Africa [18], or determining the role of swine movements in the dynamics of influenza in the United States [19]. More specifically to HIV, phylogeographic analyses have been used to reconstruct the early continental expansion of HIV-1 group M in Africa [20] and the spread of subtype B across the globe [21] and to North America [22]. However, in their study of the global spread of subtype B, Magiorkinis and colleagues noted that they were unable to use Bayesian phylogeographic methods because “the large number of geographic regions along with the higher number of sequences make the analysis to be extremely computationally intensive” [21]. This has been a common criticism of Bayesian approaches to phylodynamics, which owe most of their computational burden to accommodating phylogenetic uncertainty. While incorporating discrete traits adds to the computational burden, this can be significantly reduced by exploiting many-core computing architectures—such as multi-core central processing and graphics processing units (GPUs)—to parallelize and accelerate likelihood computations [23,24,25,26].

The limited resolution of spatial annotations in publicly available HIV sequences has also restricted phylogeographic studies to infer spatial spread at the intercontinental, international, or regional level. For HIV-1 subtype B in the United States, earlier phylodynamic studies have shown that human migration resulted in viral dispersal out of Africa and into the Americas via the Caribbean [20,22], but studies on the spatial spread within the country have been limited. In this study, we explore the ability to identify the drivers of HIV-1 subtype B dispersal in the United States by applying Bayesian phylogeographic methods to a large collection of sequences annotated at the state level. To this purpose, we adopt an extension of the standard discrete diffusion model that parameterizes relative migration rates as a function of a number of potential predictors [11]. We introduce a workflow that combines heuristic and Bayesian approaches to reduce the number of sequences and render computations feasible. By examining the consistency of the geographic reconstructions across different sample sizes and sampling schemes, we attempt to identify the limitations and compromises that are needed to perform phylogeographic reconstruction on large HIV data sets.

## 2. Materials and Methods

### 2.1. Data

We capitalize on a dataset that consists of 23,588 HIV-1 subtype B *pol* sequences collected between 2004 and 2011 from 45 different states in the United States. For a more detailed account of the data collection procedure, we refer to [27]. Briefly, the sequences were obtained from samples from HIV-1 infected individuals for HIV-1 antiretroviral resistance genotyping and generated from plasma using the ViroSeq HIV-1 Genotyping System, version 2.0 (Celera, Alameda, CA, USA), according to the manufacturer’s instructions.Sampling was heterogeneous across the states and the time frame, with the majority of sequences from the state of New York having been sampled prior to 2007 (for more details see Appendix A).

### 2.2. Subtyping

We checked for recombination and possible mislabeling of sequences by subtyping the entire dataset using the REGA HIV-1 Subtyping tool, version 3.41 [28]. Sequences with assigned subtypes that differed from ‘Subtype B’ or ‘Subtype B-like’ were removed from the analysis (n = 374/23,588). Subtype B-like assignments refer to sequences that cluster outside of the Subtype B reference clade, but are sister sequences to subtype B and fall outside of the other subtype clades. These sequences were kept in the analysis, since the references used in the REGA Subtyping tool may not capture the entire diversity of subtype B. Additionally, we removed 5 sequences that contained non-IUPAC compliant alphabets.

### 2.3. Multiple Sequence Alignment

We aligned the remaining 23,209 subtype B sequences using VIRULIGN v1.0.1 [29] to construct a multiple sequence alignment. The alignment was subsequently cleaned by removing sequences that had premature stop codons, or over 30% of unknown amino acids after translation. This procedure removed 58 sequences and produced a 1302 bp alignment of 23,161 sequences.

### 2.4. Subsampling Strategy

The magnitude of the dataset prohibits a fully Bayesian inference approach; hence, a subsampling strategy is required to enable phylogeographic analyses. A first subsampling step therefore consisted of removing sequences such that monophyletic clusters that entirely consist of samples from the same state are represented by a single sample. This is justified by the state-level geographic resolution of our data. Monophyletic clusters consisting solely of sequences from the same state do not bring any additional information on the between-state diffusion process we aim to infer. We discarded all but one randomly selected sequence per monophyletic cluster to provide a systematic way to significantly reduce the initial dataset, and kept only those sequences that pertained to our question of interest. In practice, this was done by first estimating a maximum-likelihood tree using FastTree 2.1 [30] and removing potential hypermutants (or sequences with mislabeled sampling dates) by performing a root-to-tip regression using TempEst [31]. We then constructed a new tree without the outliers, parsed the tree using the ‘ape’ R package [32] to identify the state-specific clusters, and removed the redundant sequences from their corresponding clades. Using this approach, we found a total of 3989 monophyletic clusters, which allowed us to remove 8219 out of the available 23,161 sequences (i.e., approximately 35%). Since the main objective of this step was to greatly reduce the number of sequences, we did not take into account branch support values when selecting the clusters from which we subsampled. This also allowed us to avoid setting arbitrary threshold values in the clustering step.

The resulting dataset consists of 14,938 sequences, which remains unpractical for a fully Bayesian phylodynamic inference. To render the computations manageable, we introduced a second subsampling step, in which we fixed an arbitrary total number of sequences and sampled from each US state a number proportional to the number of individuals living with HIV in each location [33]. We were unable to further stratify the sampling by year, since not all locations were represented every year. To explore the effect different sample sizes have on the phylogeographic reconstruction, we subsampled data sets of 250, 500, 750, 1000, 2500, 5000, and 10,000 sequences. If the number of available sequences in a location was lower than what would be proportional for a given sample size, the missing sequences were drawn randomly from the pool of unsampled sequences. After determining the number of sequences to sample from each state, we compared two sampling schemes: Random subsampling and subsampling based on phylogenetic diversity (PD), in order to assess the robustness of our inference. Given a phylogenetic tree, the PD of a group of taxa is defined as the sum of branch lengths of the minimal subtree spanned by those taxa [34]. To subsample using phylogenetic diversity as a criterion, we constructed a maximum-likelihood tree for each state with all available sequences from the corresponding location, and used Phylogenetic Diversity Analyzer (PDA) [35] to select the sequences that maximized PD. In contrast, random subsampling consisted of randomly drawing sequences for a given location and sample size. This resulted in a total of 14 subsamples, one for each sample size and sampling scheme.

### 2.5. Phylogenetic Inference

To further reduce computational complexity, we sidestepped the joint estimation of phylogeny and geographic history by fitting the diffusion model to pre-estimated posterior tree distributions. For each subsample of size up to 1000 taxa, we inferred an empirical distribution of time-calibrated phylogenies using BEAST v1.10.4 [36] and subsequently conditioned our phylogeographic reconstructions on these tree distributions. We modeled molecular evolution according to a GTR+$\Gamma_{4}$ [37,38] substitution model with empirical nucleotide frequencies and assumed a logistic growth population size model—with uninformative priors on the growth rate and population size parameters—following the results in [22]. The overall evolutionary rate was assumed to follow a strict molecular clock. This is a simplifying assumption in order to avoid mixing problems associated with more parameter-rich relaxed molecular clock models. As the sequence data do not contain sufficient temporal signal, we set an informative normal prior on the root height of the phylogeny, with a mean of 44 years into the past and a standard deviation of 2 years, based on recently obtained estimates [22]. The Markov chain Monte Carlo analyses in BEAST were run sufficiently long to ensure adequate chain mixing on all relevant parameters as assessed through Tracer [39], and the empirical tree distributions were constructed by sampling 1000 evenly spaced trees throughout the posterior simulation after burn-in.

For the subsample sizes of over 1000 taxa, we opted to infer approximate maximum-likelihood phylogenies using FastTree under a GTR+Γ model of molecular evolution and to time calibrate the resulting phylogenies using TreeTime v0.5.1 [40]. Subsequent phylogeographic analyses were conditioned on these (fixed-topology) time-calibrated trees after rescaling them in BEAST to match the time to the most recent common ancestor (TMRCA) calibration previously mentioned.

### 2.6. Geographic History Reconstruction

To reconstruct the spatial diffusion process, we modeled the instantaneous rate of transitions between the different states as a continuous-time Markov chain process [9]. Under this formulation, movement between *K* discrete locations is parameterized in terms of a K×K infinitesimal rate matrix Λ, where $\Lambda_{ij}$ is the instantaneous, relative transition rate from location *i* to *j*. We further informed these rates by incorporating a number *P* of potential explanatory predictors in a generalized linear model (GLM) framework [11]. The relative transition rates $\Lambda_{ij}$ are defined in terms of a log linear function of the set of these covariates, with each having a coefficient $\beta_{p}$ for p=1,…,P quantifying its contribution to Λ, and an indicator variable $\delta_{p}$ that determines their inclusion or exclusion of each predictor in the model. This model allowed us to employ Bayesian stochastic search variable selection (BSSVS) to explore the space of $2^{P}$ possible predictor combinations in the linear model, and to obtain a posterior probability on the indicator variables $\delta_{p}$ [11]. To ensure sparsity in the model and avoid overfitting, we followed [11] and assigned Bernoulli prior probability distributions on $\delta_{p}$, with success probabilities such that 50% of the prior probability mass is placed on no predictors being included. Following [41], we also included origin- and destination-specific random effects ($\epsilon_{i},\epsilon_{j}$) in our GLM specification to account for unexplained variability in the diffusion process.

We completed our model specification by assuming a priori that all $\beta_{p}$ were independent and normally distributed, with mean 0 and variance 1. We used this more restrictive prior variance compared to previous analyses to avoid numerical issues associated with extreme predictor effect sizes. To assess the impact of this change, we performed a posterior predictive test, where we infered masked taxon locations under the two prior variance specifications. The results showed that the reduced variance approach had no significant impact on the geographic reconstruction, since it produced similarly accurate results to those of the default prior settings on $\beta_{p}$. To ensure more consistent spatial reconstructions among datasets (different sizes and different subsampling), we also fixed the location of the root of each reconstruction to the state of New York. This constraint was added after noticing that assuming an unstructured shape of the tree resulted in inconsistent reconstructions of the root state, which in turn ended up having an impact on which predictors were included in the model. Fixing New York as the root state was motivated by a previous phylogeographic analysis of the introduction of HIV-1 in the US [22].

To inform the transition rates in our GLM specification, we considered a total of 38 potential predictors in each phylogeographic reconstruction. For state-specific measures (e.g., population size), we specified an origin and destination predictor that uses the measure at the origin and destination respectively as covariate value for each pairwise transition rate. Prior to log-transformation and standardization (to grant predictors equal variance a priori), we added pseudocounts for predictors that had zero entries. For a detailed description of each predictor, we refer to the Appendix A. We used BEAST v1.10.4 [36] to infer posterior estimates for all parameters, with BEAGLE v3.1.0 [26] for GPU-accelerated computation.

## 3. Results

### 3.1. Effects of Sampling Scheme on Reconstructed Tree Topologies

As expected for HIV-1 subtype phylogenies, our reconstructed topologies for all datasets have a star-like shape, regardless of the subsampling scheme used. This characteristic combination of long terminal branches and relatively short internal branches has been attributed to an underlying exponential growth in the virus population [5,42,43]. It is possible to formally assess the star-likeness of a tree by looking at the ratio between the mean terminal branch lengths and the mean internal branch lengths [44]. For smaller sample sizes (up to 1000 taxa), we observe on average that the terminal branches tend to be at least 10 times longer than the internal branches, with the random sampling scheme generating significantly different branch length ratio distributions to PDA (Kolmogorov–Smirnov test, $p<10^{-6}$, Figure 1A). We also observe that the branch length ratios on the PDA tree distributions have a smaller mean than the ones resulting from the randomly subsampled data, indicating that the trees generated using PDA sampling are, to some extent, less star-like. However, the difference in branch length ratios between sampling schemes decreases as the sample size grows and the trees become less star-like, with highly similar branch ratio values for the 2500, 5000, and 10,000 taxa trees. To check if the inference methodology (Bayesian or maximum likelihood) had a role in the sharp drop in values between the 1000 and 2500 taxa trees, we performed a similar maximum likelihood phylogenetic inference and time calibration on the 1000 taxa datasets. The branch ratios for the random and PDA subsampled trees were 7.6 and 6.9, respectively. Although the PDA tree remains less star-like, the overall lower values when compared to the Bayesian results show that the inference method being used does play a role.

Moreover, the variance of the branch ratio distributions is larger in the randomly sampled datasets, implying a higher uncertainty in the branch lengths reconstructed, but not necessarily saying anything on the variability in topology. A formal way to examine this variability in topologies is by comparing the maximum clade credibility (MCC) scores of each tree distribution. We used TreeAnnotator to find the highest log clade credibility score—the log product of posterior probability support values for each node in the maximum clade credibility tree—and tree for each distribution (Figure 1B). For all sample sizes, the datasets sampled with PDA are associated with higher log(MCC) scores, which reflects less uncertainty and a more constrained set of topologies in the tree distribution.

### 3.2. Phylogeographic Reconstruction and Drivers of Spread

The spatial process reconstructed by our phylogeographic analysis suggests that the patterns of HIV-1B spread in the US are characterized by source–sink dynamics. This pattern of spread is defined by the fact that the majority of the movement between locations happens from source to sink states, without any intermediate transitions through other states. We can visually inspect this by means of connectivity plots, where the transitions between locations are extracted from the MCC trees and projected into geographic space, with lines connecting the two locations and circles indicating the origin of each transition.

The connectivity plots show that this source–sink pattern is most pronounced in the smaller subsample sizes—up to 1000 taxa for random subsampling (Figure 2) and 750 for PDA (see Appendix A)—where the geographic reconstructions show all transitions originating from the state of New York to other states. As the number of sequences grows beyond these subsample sizes, the source–sink dynamics remain, but with the number of sources increasing to include the states of California and occasionally Florida and Texas. Interestingly, for the randomly subsampled 10,000 taxa dataset, we see a considerable increase in the connectivity of the spatial process, with transitions originating from over 15 different locations.

The connectivity plots show that this source–sink pattern is most pronounced in the smaller subsample sizes—up to 1000 taxa for random subsampling (Figure 2) and 750 for PDA (see Appendix A)—where the geographic reconstructions show all transitions originating from the state of New York to other states. As the number of sequences grows beyond these subsample sizes, the source–sink dynamics remain, but with the number of sources increasing to include the states of California and occasionally Florida and Texas. Interestingly, for the randomly subsampled 10,000 taxa dataset, we see a considerable increase in the connectivity of the spatial process, with transitions originating from over 15 different locations.

In line with the sparse source–sink connectivity patterns, our phylogeographic reconstructions generally associate only a few of the potential predictors of HIV-1B spread in the US to viral lineage movement. The mean number of predictors included in the model ranges from two to eight, a small fraction of the total of 38 potential predictors (Appendix A). Given that we only observe a single realization of the spatial process (a single ‘site’ of discrete location states at the tree tips), sparseness in the number of explanatory variables in the model is a desired property encoded in our prior specification that offers protection against over-fitting. The mean number of predictors included tends to grow as the sample size becomes larger, ranging from 2–6 for the random subsampling, and 2–8 for PDA (Appendix A). This increase can be attributed to the larger number of transition events in the ancestral history that need be explained, as reflected in the connectivity plots (Figure 2). Interestingly, we also observe that the number of predictors never included in the model increases in the larger subsample sizes (Appendix A). This potentially indicates a reduction of uncertainty with increasing data, but we are not able to decouple this effect from the fact that the larger subsample analyses are conditioned on only a single tree topology.

We are able to consistently recover three well-supported predictors across the different analyses (Table 1): Population size at origin (Population Origin), population size at destination (Population Destination), and income inequality at origin (Gini Origin). All of these predictors are included in 10 out of the 14 subsamples. Population size at destination and income inequality at the origin appear highly supported with a Bayes factor (BF) >30 in all included instances under each sampling scheme. In contrast, the support for the population size at origin is less consistent, with a BF above 30 in only 5 out of 10 instances. The mean conditional effect size of these predictors was positive in all cases, implying that transition rates between origin and destination states increase with larger values for these variables. Despite the consistency in these predictors, we find it difficult to interpret how the variables relate to the actual epidemic process, since the lack of connectivity we see in many of the reconstructions suggests that the predictors only reflect characteristics of the source–sink dynamics mainly originating in the state of New York. Moreover, we observe that the predictors included in the 10,000 taxa subsamples differ from those in the other datasets (Figure 3). This is consistent with the increase in connectivity that we infer in the corresponding geographical reconstructions (Figure 2). In these analyses, support for interesting mobility-related predictors emerges, such as the number of passengers on flights between each pair of locations (Aviation, PDA and random) and commuting flows (Commuting, only in PDA), or the geographic adjacency between two states (Adjacency, PDA and random) that could serve as a proxy for commuting or other short-range mobility. However, it remains challenging to draw clear conclusions about this because, particularly for PDA, an unrealistically large number of predictors receive very high support.

Other predictors supported in at least one of the analyses with a Bayes Factor greater than 5 (Figure 3) include: The percentage of the population that is black at destination (Black Destination), the percentage of high schools in origin states with mandatory sex education (Condom Origin), geographic distances (Distance), ecological distances in terms of inaccessibility relative to travel time required to reach the nearest urban center (Inaccessibility), heroin mortality rates at destination (Heroin Destination), the proportion of men who have sex with men at origin (MSM Origin), and measures of sampling bias—included to absorb the spurious inclusion of predictors due to unevenness in sampling—such as the number of samples in each state (Samples Origin and Destination) and the residuals for origin states (Residuals Origin) obtained from a linear regression of the number of sequences sampled from each location, against the average number of individuals living with HIV. Like with the more consistently included predictors, we advise caution at the time of interpretation, since the inclusion of these predictors may be the result of a biased phylogeographic reconstruction. For a detailed description of how all predictors were obtained, see Appendix A.

## 4. Discussion

In our attempt to identify the factors that shaped the spatial history of HIV-1 subtype B in the US using large numbers of sequences, we applied a combination of heuristic and Bayesian phylogeographic methods to reconstruct the historical dispersal patterns of the virus since its introduction into the country. Similar hybrid approaches have been shown to yield reasonable results when inferring demographic parameters on bacterial data [45], provided that the data exhibit sufficient temporal signal. While our study is not able to successfully reconstruct a detailed epidemic history of HIV-1B in the US, it highlights the limitations and challenges when performing phylogeographic inference with this virus.

To overcome the computational demands involved in jointly inferring phylogenies and spatial spread, we conditioned the analyses on empirical tree distributions and maximum-likelihood phylogenies to reconstruct the ancestral locations along the trees. This hybrid approach yielded consistent results under reasonable run times, but even on the fixed trees using GPUs [26], the high-dimensional state spaces of geographical locations on trees made scaling the analysis to larger numbers of sequences difficult. To avoid numerical issues and overfitting, we incorporated random effects into our GLM model and employed a low prior variance on the GLM coefficients.

Moreover, the temporal signal in our collection of sequences was found to be weak, which required us to incorporate external information to calibrate branch lengths in units of time. The origins of HIV-1 in the US have been extensively studied and documented [22,46], which enabled us to make well-informed choices about model and prior specification, but we acknowledge that this approach might not be generalizable to other less-well-studied pathogens. Nevertheless, incorporating information from previous studies through prior specification fits well within the philosophy of Bayesian inference.

The fundamental challenge we encountered was biological in nature and was related to the evolutionary history of the virus. Previous molecular dating work places the introduction of HIV-1 into the US in the late 1960s in the state of New York, upon which the virus rapidly spread throughout the country, as also reflected in a fast exponential growth phase [22,46]. The phylogenies generated under this demographic process result in star-like shapes [3,42,43]. This provides little opportunity to identify structured relatedness among samples from different locations, and it therefore remains difficult to adequately inform the parameters of spatial transitioning, resulting in a high degree of uncertainty in the reconstructions. Adding more sequences to the data set produced less star-like phylogenies, but this was associated with increased computational demands.

For the data sets including up to 1000 sequences, the trees we obtained are not very informative, and we are unable to learn anything relevant about the underlying epidemic process, since the geographical reconstructions conditioned on these trees result in extreme source–sink scenarios with predictors of spatial spread reflecting mostly characteristics of the source location (at the root of the phylogenies). We are therefore unable to uncover many plausible transitions between intermediate states along the long external branches of the star-like topologies. For the larger datasets, the maximum-likelihood trees are apparently more informative, but the results from the spatial process reconstruction are likely less generalizable, as they are obtained by conditioning on a single tree topology. The use of fixed tree topologies on the larger datasets is unable to capture the considerable uncertainty present when reconstructing star-like trees, which demands caution when interpreting the predictors included in the model, as they might not necessarily capture the true epidemiology of the virus and instead only offer a phenomenological description of a process on a single topology in question.

Our analyses were further limited by the fact that we are only able to test for potential predictors that are assumed to be fixed throughout time, and the fact that we can only consider spatial resolution at the state level, which might be too coarse to fully capture the underlying epidemic given the size of and socio-economic diversity within each state. A city or county level would allow the inferring of more detailed spatial dynamics, but at the cost of a dramatic increase in the geographic state space. An alternative method that would avoid modeling diffusion on large phylogenies could consist of focusing on tracing individual transmission chains and testing statistical associations of potential predictors with the detected transmission clusters. HIV-TRACE [47] is a tool that offers scalable ways for rapid detection of transmission clusters using molecular data, which is now routinely used by public health practitioners in the US [48]. HIV-TRACE is fundamentally different from phylogenetic approaches since its focus is not in inferring shared ancestry, but instead tries to identify chains of partners whose viral genetic relatedness imply direct or indirect epidemiological connections by computing a large number of pairwise genetic distances under a clustering threshold [47]. By characterizing every individual within each transmission cluster, one could, in theory, pool the data of different transmission chains to test the association of explanatory variables with cluster composition. However, this method would require large amounts of sensitive data on each individual associated with the network, which limits its ease of implementation. Additionally, it does not offer ways to explicitly infer the directionality of transmission or examine which variables drive cluster formation and growth. However, complementary to ancestral reconstruction on star-like trees, such an approach would be able to recover more recent dispersal dynamics.

Overall, the limitations we acknowledge here indicate that currently available methods fall short for jointly estimating the spatial and evolutionary process underlying the HIV-1B epidemic in the US. A major limitation we observed was the inherent non-informative nature of trees from this pathogen. Perhaps a more insightful study would involve examining the spatial spread patterns of non-B subtypes in the US. Although subtype B is predominant in the US, non-subtype B infections have been detected from all regions across the country [27] and have been linked to a variety of factors, including immigration, travel, military deployment, etc. If the different subtypes follow a similar spatial process, we speculate that performing a similar analysis pooling the information from such subtypes may yield more informative results in terms of predictors of dispersal, given their shorter evolutionary and transmission histories.

In conclusion, the workflow we provide offers ways to confront the challenges encountered when performing Bayesian phylogeographic inference on large datasets. Most notably, we introduce novel subsampling strategies to reduce the number of taxa to more manageable numbers. Prior studies have independently used phylogenetic diversity and geographic clusters with the purpose of accounting for sampling biases and reducing the number of sequences [15,16,17], but have not applied these methods to phylogeographic analyses of this magnitude. Our phylogenetically informed subsampling procedure offers a systematic and objective way to significantly reduce the initial number of sequences while preserving as much information on the underlying spatial diffusion process as possible. Further subsampling using epidemiological data was done to complete the analysis. This two-step subsampling approach has advantages over blindly sampling sequences according to prevalence, as it allows us to avoid oversampling from a transmission chain localized within a single state. We note that the limitations we encountered are not necessarily predictive of the applicability of the methodology to HIV-1 in other geographic settings or to other viruses on this geographic scale. Seasonal influenza, for example, is characterized by different circulation patterns and evolutionary dynamics, with repeated introductions every season and with ladder-like tree structures. The procedures we propose may therefore find interesting applications for the large amounts of genomic data that have become available for these and other pathogens at various geographic scales.

## Figures and Tables

**Figure 1 viruses-12-00182-f001:**
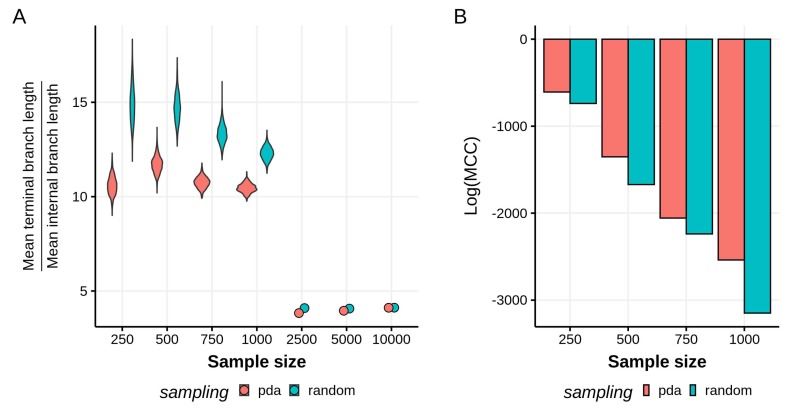
(**A**) Ratios between the mean terminal branch lengths and the mean internal branch lengths. The higher this ratio, the more star-like a phylogeny. Violin plots are shown for sample sizes in which tree distributions are available. (**B**) Log-maximum clade credibility (MCC) scores of each empirical tree distribution. Tree distributions sampled using the Phylogenetic Diversity Analyzer (PDA) method have higher MCC scores, implying somewhat less variance in the tree topologies reconstructed.

**Figure 2 viruses-12-00182-f002:**
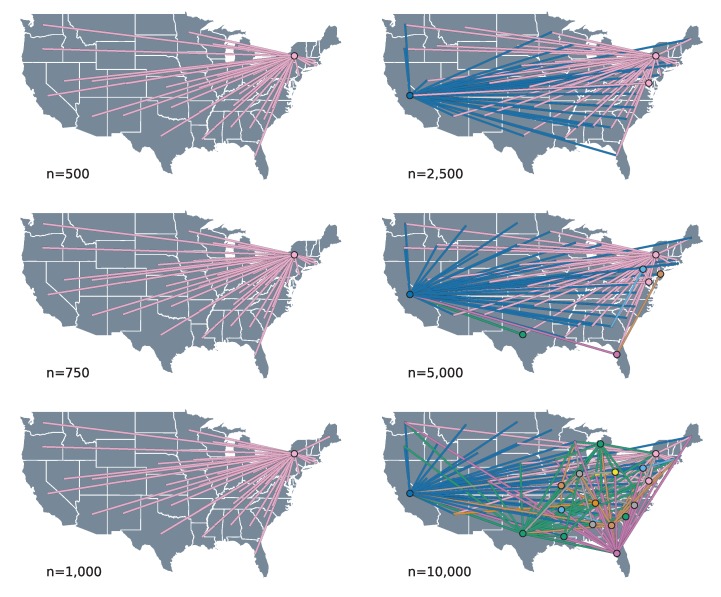
Connectivity plots for the randomly subsampled datasets. The geographical reconstructions from each MCC tree are projected into geographical space by drawing a line for each transition event, connecting origin (circle), and destination. The connectivity plot for the 250 taxa tree is not included in this plot because it was found to be similar to those of the 500, 750, and 1000 taxa datasets. For the connectivity plots for the datasets subsampled using the Phylogenetic Diversity Analyzer, we refer to the Appendix A.

**Figure 3 viruses-12-00182-f003:**
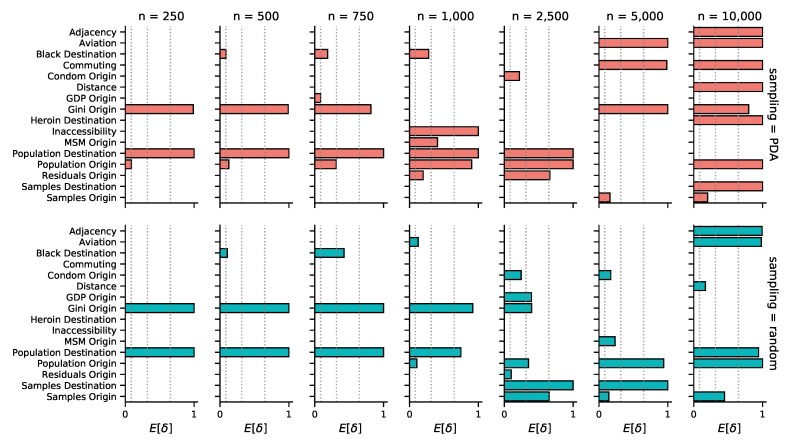
Predictors of HIV-1B spatial spread within the United States by subsample size and sampling scheme (PDA/random). Posterior inclusion probabilities for each predictor are represented in terms of indicator expectations (E[δ]). Only predictors with a Bayes factor greater than 5 in at least one analysis are displayed. Dotted vertical lines denote inclusion probabilities corresponding to Bayes factors of 5, 25, and 100, respectively. For all inclusion probabilities and a description of each predictor, we refer to Appendix A.

**Table 1 viruses-12-00182-t001:** Predictors included in over 50% of subsamples

Predictor	Times Included (BF>5)	Bayes Factor ${}^{1}$	Coefficient ${}^{2}$
Population size at origin	10/14	265.4 (5.2 - *∞*)	2.8 (1.4 - 3.5)
Population size at destination	10/14	*∞* (160.1 - *∞*)	1.0 (0.8 - 1.6)
Gini index at origin ${}^{3}$	10/14	5165.4 (36.0 - *∞*)	2.7 (1.0 - 4.2)

^1^ Median, minimum, and maximum Bayes Factors across all subsamples in which the predictor was included. ^2^ Median, minimum, and maximum mean predictor coefficient (conditioned on inclusion, E[β|δ=1]). ^3^ Income inequality Gini coefficient.

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
