# Peer review of "In Search of Covariates of HIV-1 Subtype B Spread in the United States—A Cautionary Tale of Large-Scale Bayesian Phylogeography"

_viruses, 2020, doi:10.3390/v12020182_

Round 1

Reviewer 1 Report

In this work the authors provide a large phylogenetic analysis of covariates associated with HIV-1 subtype B spread in the United States. Due to the large scale of their sequence databases and also to various bias (different time and geographic coverage quality), they failed to obtain strong data regarding those covariates. Thus, the authors are not discussing the factors of HIV subtype B spread in the US, but are providing and discussing in details the phylogenetic framework used. As Bayesian phylogeography is computationally intensive, it can usually not be used on large sequence database with a large number of estimates. Here, the authors used a complex framework to allow the Bayesian framework use and to check all the potential biases. It is a very well explained and discussed process that could be apply, with success, to other datasets. All the potential limitations due to large calculation and sampling biases are very well explained, step by step, and also illustrate all the pitfalls of such analysis that should always be taken into account when interpreting such results.

This work is very informative about the minimal number of dataset to be included and the increasing complexity of such covariate analysis over large time period when both the number and the types of populations concerned by the disease is greatly increasing.

The authors are some of the worldwide masters of such phylogenetic frameworks. The manuscript is clear and very well written, the main steps of the process are very well explained and all potential biases largely discussed. The factors identified with subtype B spread are still presented but with adequate caution and warnings.

The article deserve publication as it is in my opinion and I have only two, really minor, comments.

Figure 3. The predictors’ names and acronyms are not always easy to decipher. They should be given in a figure footnote and/or in the text of the manuscript at the time of their discussion. Line 193. The figure 1A does not present the trees’ shapes. This is no big deal and I understand why the authors cited it there. But, as the link between branch length ratio and star-shape trees is explained a few sentence later with the reference to the figure, I would discard this reference from line 193.

Author Response

We would like to thank the reviewer for their assessment of our manuscript and the comments provided. Below you will find a response to the edits suggested.

Comment 1Figure 3. The predictors’ names and acronyms are not always easy to decipher. They should be given in a figure footnote and/or in the text of the manuscript at the time of their discussion.

Response 1: Following the suggestions of the reviewer, we have expanded the Results section of the manuscript to include a brief description of each predictor name introduced in Figure 3.

Comment 2Line 193. The figure 1A does not present the trees’ shapes. This is no big deal and I understand why the authors cited it there. But, as the link between branch length ratio and star-shape trees is explained a few sentence later with the reference to the figure, I would discard this reference from line 193. 

Response 2: We agree with the comment of the reviewer and have removed the Figure 1 reference as suggested. 

Once again, we thank the reviewer for the time and effort put in the assessment of our manuscript.

Reviewer 2 Report

none the authors well applied all the principles described in the paper

Author Response

We would like to thank the reviewer for their assessment of our manuscript.